# Smad2/3 Activation Regulates Smad1/5/8 Signaling via a Negative Feedback Loop to Inhibit 3T3-L1 Adipogenesis

**DOI:** 10.3390/ijms22168472

**Published:** 2021-08-06

**Authors:** Senem Aykul, Jordan Maust, Vijayalakshmi Thamilselvan, Monique Floer, Erik Martinez-Hackert

**Affiliations:** 1Department of Biochemistry and Molecular Biology, Michigan State University, 603 Wilson Road, East Lansing, MI 48824, USA; senem.aykul@regeneron.com (S.A.); maustjor@msu.edu (J.M.); thamilse@msu.edu (V.T.); floer@advertentbio.com (M.F.); 2Regeneron Pharma, Tarrytown, NY 10591, USA; 3Advertent Biotherapeutics, East Lansing, MI 48824, USA

**Keywords:** adipogenesis, signaling, TGF-β, BMP, SMAD, 3T3-L1

## Abstract

Adipose tissues (AT) expand in response to energy surplus through adipocyte hypertrophy and hyperplasia. The latter, also known as adipogenesis, is a process by which multipotent precursors differentiate to form mature adipocytes. This process is directed by developmental cues that include members of the TGF-β family. Our goal here was to elucidate, using the 3T3-L1 adipogenesis model, how TGF-β family growth factors and inhibitors regulate adipocyte development. We show that ligands of the Activin and TGF-β families, several ligand traps, and the SMAD1/5/8 signaling inhibitor LDN-193189 profoundly suppressed 3T3-L1 adipogenesis. Strikingly, anti-adipogenic traps and ligands engaged the same mechanism of action involving the simultaneous activation of SMAD2/3 and inhibition of SMAD1/5/8 signaling. This effect was rescued by the SMAD2/3 signaling inhibitor SB-431542. By contrast, although LDN-193189 also suppressed SMAD1/5/8 signaling and adipogenesis, its effect could not be rescued by SB-431542. Collectively, these findings reveal the fundamental role of SMAD1/5/8 for 3T3-L1 adipogenesis, and potentially identify a negative feedback loop that links SMAD2/3 activation with SMAD1/5/8 inhibition in adipogenic precursors.

## 1. Introduction

Adipose tissues (ATs) are essential for regulating energy balance and for maintaining metabolic, endocrine, and immune health [1]. In obesity, which is characterized by the massive expansion of AT driven by disproportionately high energy intake relative to energy expenditure, the regulating functions of AT can become severely compromised [2]. The underlying cluster of conditions associated with AT dysfunction are collectively known as Metabolic Syndrome (MetS) and include increased blood pressure, hyperglycemia, and elevated cholesterol or triglyceride levels. These conditions frequently lead to serious co-morbidities such as type 2 diabetes, cardiovascular disease, nonalcoholic steatohepatitis, and cancers [3], making the current obesity epidemic one of the most pressing public health challenges of our time [4].

AT expansion in response caloric excess is driven by two distinct cellular processes: adipocyte hypertrophy and hyperplasia (or adipogenesis) [1]. In hypertrophy, existing adipocytes become enlarged by storing excess energy as triglycerides. In hyperplasia, ATs expand as resident precursors proliferate and differentiate into mature adipocytes [5,6,7]. Hyperplastic AT expansion in adults occurs mainly in the intra-abdominal region (i.e., visceral AT, or VAT), in skeletal muscle, and in bone marrow [7,8,9,10,11]. Notably, increased adiposity at these sites associates with poor metabolic health and other obesity related co-morbidities [11,12,13,14,15]. Elucidating the mechanisms and identifying the factors that regulate AT expansion is, therefore, critical from a public health point of view and could help in the development of new therapeutic strategies and targets for treating obesity and its associated co-morbidities.

Transforming Growth Factor-β (TGF-β) ligands, which include TGF-βs, Activins, GDFs, and BMPs, have critical regulating functions in adipocyte hypertrophy and hyperplasia or adipogenesis [16,17,18]. For example, TGF-β1, GDF-8, GDF-11, Activin A, and Activin B have been shown to inhibit adipogenesis, while several BMPs have been shown to promote adipogenesis of pre-adipocyte-like cell lines or primary cultures [19,20,21,22,23,24,25,26,27,28,29,30,31,32]. At a molecular level, these ligands exert their function by forming a signaling complex with two ‘type I’ and two ‘type II’ receptors [33], thus initiating a signal transduction cascade that involves phosphorylation of R-SMAD transcription factors at their C-terminal serine residues [34], hetero-oligomerization of phosphorylated R-SMADs with the co-transcription factor SMAD4, and translocation of the phospho-R-SMAD-SMAD4 complex to the nucleus to regulate gene expression. R-SMADs can be divided into the SMAD2/3 branch, which is activated by TGF-βs and Activins together with the type I receptors ALK4, ALK5, or ALK7, and the SMAD1/5/8 branch, which is activated by BMPs, most GDFs and the type I receptors ALK1, ALK2, ALK3 and ALK6 [34]. TGF-β family ligands can also activate non-SMAD pathways, including extracellular signal-regulated kinases (ERK), c-Jun amino terminal kinase (JNK), p38 MAPK, TAK1 and others [35,36,37]. Although much work aimed at understanding the roles of the TGF-β family in adipogenesis has been published, fundamental questions remain, including what roles ligands play in different aspects of adipogenesis, how intracellular signaling pathways interact to regulate adipocyte development, and whether TGF-β family inhibitors can suppress adipogenesis. 

To address these questions, we investigated the roles of 11 TGF-β family ligands, 11 inhibitory ligand traps, and 2 small molecule signaling inhibitors in adipocyte precursor commitment, proliferation, and adipocyte hypertrophy using the 3T3-L1 adipogenesis model. Confirming earlier results, we found that TGF-β-like ligands, which primarily activate intracellular SMAD2/3 pathways, suppressed pre-adipocyte differentiation. By contrast, BMP-like ligands, which primarily activate SMAD1/5/8 pathways, promoted an increase in adipocyte number. Three ligand traps and the SMAD1/5/8 signaling inhibitor LDN-193189 suppressed adipogenesis. Strikingly, both anti-adipogenic traps and ligands exhibited the same mechanism of regulation, which consisted of simultaneous SMAD2/3 pathway activation and SMAD1/5/8 pathway inhibition. Significantly, the SMAD2/3 signaling inhibitor SB-431542 rescued adipogenesis and SMAD1/5/8 signaling in the presence of anti-adipogenic traps or ligands, but not in cells treated with LDN-193189. Our findings, therefore, indicate that SMAD1/5/8 signaling is fundamental for priming and driving commitment of 3T3-L1 cells toward adipogenic fates, whereas SMAD2/3 activation may blunt adipogenesis via a negative feedback loop that reduces SMAD1/5/8 signaling. Lastly, we identify three ligand traps that inhibit adipogenesis and, therefore, could be used to regulate hyperplastic AT expansion.

## 2. Results

### 2.1. TGF-β Ligands Differentially Affect 3T3-L1 Adipogenesis

TGF-β pathways are known to regulate adipogenic fates [18]. To identify members of the TGF-β family that contribute to 3T3-L1 adipogenesis, we analyzed their expression in differentiating 3T3-L1 cells using publicly available microarray data [38] and a web-based genomics analysis platform [39] (Appendix A). Timepoints in this study were taken at days −2, 0, 2 and 7 of differentiation [38] (Figure 1A). We found the type I receptors ALK2, ALK3, ALK4 and ALK5, and the type II receptors ActRIIA, TGFβRII and BMPRII to be expressed in 3T3-L1 cells at all timepoints. Among co-receptors, only betaglycan (a.k.a. TGFβR3) was detected, highlighting its potential role in adipogenesis [26]. The SMAD2/3 pathway activating ligands TGF-β1, TGF-β2, TGF-β3, Activin A and Activin B were also detected. This finding was surprising, as their exogeneous addition blunted adipogenesis. TGF-β3 and Activin A levels were reduced with differentiation. Activin B levels also decreased at the beginning of differentiation. However, they increased significantly as adipocytes reached maturity. Notably, Activin B can activate both SMAD2/3 and SMAD1/5/8 signaling. Among SMAD1/5/8 pathway activating ligands, GDF5 was most significantly induced with differentiation, supporting the conclusion that it may be one of the endogenous ligands that drive adipogenesis [40]. HUGO gene names for all members of the pathway are listed in Appendix A. Other studies not discussed here support these conclusions.

To identify steps controlled by TGF-β pathways in adipogenesis, we investigated the time-dependent effect of 11 different ligands on 3T3-L1 differentiation following a standard 3T3-L1 differentiation assay [41] (Figure 1A). We found that TGF-β-like ligands (i.e., TGF-βs, Activins, GDF-8 and GDF-11, which mainly activate SMAD2/3 pathways via the receptor kinases ALK4, ALK5, or ALK7 [42,43]), inhibited 3T3-L1 adipogenesis, as indicated by the near complete absence of lipid droplet (LD) formation in samples treated from the beginning of differentiation (day 0) until harvest (day 8) (Figure 1B,C). This effect was attenuated but still evident when cells were treated from later stages of differentiation (day 5) until harvest (day 8). Intriguingly, the few LDs that formed in treated samples appeared to be enlarged (Figure 1D), indicating that TGF-β-like ligands could promote hypertrophy of mature adipocytes. In addition, treated samples had fewer nuclei per well than untreated controls (Figure 1E), indicating that TGF-β-like ligands may limit proliferation, or reduce pre-adipocyte number through another mechanism. Although GDF-8 appeared to be an exception in this group, higher concentrations may be required due to its lower potency [44,45].

In contrast to TGF-β-like ligands, BMP-like ligands (i.e., BMPs and most GDFs, which activate SMAD1/5/8 pathways via the receptor kinases ALK1, ALK2, ALK3, or ALK6 [42,43]) mostly promoted 3T3-L1 adipocyte number expansion, as indicated by the greater number of nuclei per well relative to untreated controls (Figure 2A–D). However, their activities were somewhat divergent. For example, BMP-2, BMP-4, BMP-6, and BMP-7, which signal via the kinases ALK2, ALK3, or ALK6 [46], promoted proliferation with varying degrees of potency (Figure 2D). By contrast, BMP-9 prevented differentiation and BMP-10 promoted expansion (Figure 2B,D). While both ligands are known to signal predominantly via the receptor kinase ALK1 [47,48,49,50,51,52], a potentially critical difference may be that BMP-9 may also signal via ALK2, whereas BMP-10 may also signal via ALK3 and ALK6. Notably, BMP-like ligands did not significantly stimulate lipid accumulation, as evidenced by the stable number of lipid droplets and lipid droplet size in treated and control samples (Figure 2B,C). Collectively, these findings indicate that BMP-like ligands that signal via the kinases ALK2, ALK3, or ALK6 promote adipocyte formation and possibly proliferation.

### 2.2. Screen Identifies Ligand Traps with Anti-Adipogenic Activity

To discover compounds that modulate adipogenic fates by inhibiting TGF-β family ligands, we investigated how 11 Fc-fusion traps that inhibit different groups of TGF-β family ligands affect 3T3-L1 differentiation. Traps were based on the ligand binding domains of TGF-β family type I receptors (ALK2, ALK3, ALK4), type II receptors (ActRIIA, ActRIIB, TGFβRII, BMPRII), antagonists (Cerberus) and co-receptors (mCryptic, Cripto-1, BAMBI) (Figure 3). Their ligand binding activities are summarized in Appendix A [53,54,55,56,57]. Of the tested traps, TGFβRII-Fc, mCryptic-Fc, and Cripto-1-Fc profoundly suppressed adipocyte formation from precursors, as indicated by the near total absence of LDs in cells treated at the beginning of differentiation (Figure 3). Matching our ligands results, the few cells that differentiated in the presence of treatment had enlarged LDs (Figure 3C). In addition, inhibitory traps reduced the number of adipocytes, as treated samples had fewer nuclei per well (Figure 3B,D).

Confluent 3T3-L1 cells are believed to be post mitotic, yet we detected variability in the number of nuclei with different treatments. We therefore hypothesized that certain treatments could alter proliferation rates or survival of differentiating 3T3-L1 cells. Using a cell viability assay, we detected a statistically significant increase in cell number at 24 h of differentiation with BMP-6 treatment (Appendix A). Similarly, we saw increased BrdU incorporation at 72 h and increased MTT activity at 24 h in BMP-6 treated cells relative to TGF-β1 and TGFβRII-Fc treated cells (Appendix A). By contrast, we saw increased apoptosis in TGF-β1, and TGFβRII-Fc treated cells relative to untreated or BMP-6 treated samples (Appendix A). Together with our previous findings on the number of nuclei per well, these results indicate that TGF-β1 and TGFβRII-Fc may suppress proliferation and promote apoptosis, whereas BMP-6 may promote proliferation and protect against apoptosis.

### 2.3. Inhibitory Traps Suppress Adipogenic Gene Expression

To link the phenotype elicited by the inhibitory traps with adipogenesis, we investigated using qRT-PCR their effect on adipogenic gene expression as previously described (Figure 4) [58]. We found that the inhibitory traps mCryptic-Fc and Cripto-1-Fc suppressed expression of adipogenic master regulators and other adipogenesis associated transcription factors, including peroxisome proliferator-activated receptor-γ and -δ (*Pparg* and *Ppard*) [59]. Although *Pparg* was induced 10.5-fold in untreated samples relative to undifferentiated controls, its expression was not induced in cells treated with mCryptic-Fc or Cripto-1-Fc (Figure 4A). Similarly, mRNA levels of the adipocyte lineage specific transcription factor C/EBPβ increased approximately 2.5-fold in differentiated control samples but were unchanged in treated samples (Figure 4B). In addition to adipogenic master regulators, both mCryptic-Fc and Cripto-1-Fc suppressed expression of adipocyte marker genes, including fatty acid binding protein 4 (*Fabp4*), cell death-inducing DFFA-like effectors a and c (*Cidea* and *Cidec*), perilipin (*Plin1*), adiponectin (*Adipoq*) and others (Figure 4C–H). For example, levels of the adipokine Adiponectin (*Adipoq*) increased about 400-fold in control samples but only 10-fold in treated samples (Figure 4C). Fatty acid-binding protein 4 (*Fabp4)* expression increased approximately 150-fold with differentiation in control samples but only about 2- to 4-fold in mCryptic-Fc or Cripto-1-Fc treated samples (Figure 4D). More strikingly, mRNA levels of *Cidec,* a regulator of adipocyte lipid metabolism that binds to lipid droplets and regulates their enlargement to restrict lipolysis and favor storage, increased over 5500-fold after differentiation in control samples but only 44- to 70-fold in treated samples (Figure 4E). Similarly, mRNA levels *Plin1*, a gene encoding for the lipid droplet-associated protein Perilipin, were increased approximately 970-fold in the control samples but levels in the mCryptic-Fc and Cripto-1-Fc treated samples only increased about 10-fold (Figure 4G). Collectively, these findings indicate that traps and ligands that prevent lipid droplet formation block adipogenesis.

### 2.4. Anti-Adipogenic Ligands and Traps Alter SMAD1/5/8 Phosphorylation States

As SMAD transcription factors mediate intracellular responses to TGF-β family signals, we examined their C-terminal phosphorylation, which represents the activated state (Figure 5A–C). Using an anti-p-SMAD1/5/8 monoclonal antibody, we observed a strong band of approximately 50 kDa at the beginning of differentiation (day 0). This finding highlights the importance of SMAD1/5/8 signaling during early stages of adipogenesis. The 50 kDa p-SMAD1/5/8 band became considerably weaker after day 5 but persisted in all samples that differentiated (Figure 5A,B), indicating that SMAD1/5/8 signaling may play a lesser role in mature adipocytes. Strikingly, the 50 kDa p-SMAD1/5/8 band was almost completely superseded by a new, higher molecular weight species of 55–60 kDa in all cells treated with traps or ligands that arrested differentiation (Figure 5A,B, upper panel). For simplicity, we refer hereafter to the 50 and 55–60 kDa bands as p-SMAD1/5^Lo^ and p-SMAD1/5^Hi^. In contrast to p-SMAD1/5/8, we only observed weak 50 and 55–60 kDa bands with the p-SMAD2/3 monoclonal antibody in untreated cells at days 3 and 5 of differentiation (Figure 5A, middle panel). These findings indicate that SMAD2/3 signaling is not activated during adipogenesis. However, cells treated with inhibitory or non-inhibitory traps or ligands presented a stronger p-SMAD2/3 band at day 3 of differentiation (Figure 5A,B, middle panel).

To establish the identity of the distinct p-SMAD bands, we probed samples with antibodies against unique sequences within the SMAD linker regions (Figure 5C). A SMAD1 specific monoclonal antibody reacted both with the p-SMAD1/5^Lo^ and p-SMAD1/5^Hi^ bands. By contrast, a SMAD5 monoclonal and a SMAD8 polyclonal antibody only reacted with the lower molecular weight form. These results are consistent with earlier knock-down data identifying SMAD1 and SMAD5 forms of distinct molecular weights and suggest that SMAD1 likely is the major SMAD associated with the higher molecular weight species [60]. Similarly, a SMAD3 specific monoclonal antibody reacted with the p-SMAD2/3 band and a lower molecular weight band, while a SMAD2 specific monoclonal antibody only reacted with the lower molecular weight band. Although the molecular weight of SMAD3 is less than that of SMADs -1, -2, -5, and -8 (i.e., 48.1 kDa compared with approximately 52.3 kDa), these findings support the previous conclusion that SMAD3 is activated by adipogenesis inhibitors [61]. To corroborate that SMAD1/5 is activated in adipogenesis, we immunoprecipitated (IP’d) cell lysate from differentiating cells using anti-p-SMAD1/5/8 and probed this sample with linker specific antibodies (Figure 5D). SMAD1 and SMAD5 antibodies reacted with the IP’d sample, whereas SMAD2 or SMAD3 antibodies did not. Thus, these immunoblot-based results indicate that SMAD1/5 activation is associated with adipogenesis, whereas SMAD3 activation is associated with adipogenesis arrest.

As SMADs have multiple phosphorylation sites in addition to their C-terminal Serines (Appendix A), and as SMAD linker- or hyper-phosphorylation has been widely reported to regulate signaling activities [62,63,64,65,66,67], we hypothesized p-SMAD1/5^Hi^ could represent a hyper-phosphorylated form. To test this hypothesis, we treated lysates of control and TGFβRII-Fc treated samples with alkaline phosphatase (AP) and evaluated changes in SMAD1/5 electrophoretic mobility using different antibodies (Figure 5E). AP treated samples were largely undetectable by the p-SMAD1/5/8 antibody due to near complete loss of all phosphate groups. By contrast, SMAD1 and SMAD5 antibodies reacted well with both untreated and AP treated samples. We saw a small difference in electrophoretic mobility in control samples following AP digestion and mainly detected a band that corresponded to p-SMAD1/5^Lo^. Strikingly, the band corresponding to p-SMAD1/5^Hi^ in TGFβRII-Fc (TFc) treated samples was reduced by AP treatment to an electrophoretic mobility that corresponded to p-SMAD1/5^Lo^. These results indicate that p-SMAD1/5^Hi^ could represent a hyper-phosphorylated SMAD1/5. However, we cannot entirely rule out other SMAD/1/5 post-translational modifications or antibody cross-reactivity with p-SMAD2/3.

Several kinases, including ERK, JNK and PI3K, are thought to hyper-phosphorylate SMADs [62,63,64,65,66,67,68,69]. To determine if any one of these kinases engages in SMAD1/5/8 hyper-phosphorylation, we investigated their activation state using the phospho-kinase profiler array kit (Appendix A). All tested samples, including undifferentiated precursors and differentiating adipocytes, showed significant GSK-3α/β and WNK1 phosphorylation, suggesting that these kinases may have relevant roles in adipogenesis. However, these results also indicate that GSK-3α/β and WNK1 may not be involved in 3T3-L1 adipogenesis arrest, as their activation levels are unchanged in treated cells. By contrast, ERK phosphorylation was increased in cells treated with adipogenesis inhibitors. However, further analysis did not provide evidence that ERK gives rise to p-SMAD1/5^Hi^. Thus, a still undefined kinase may be responsible for the proposed, shift-inducing phosphorylation.

### 2.5. Adipogenesis Inhibitors Suppress SMAD1/5/8 Signaling

Hyper-phosphorylated SMADs are known to exhibit altered cellular localization and signaling [62,63,64,65]. To establish how the different p-SMAD forms and treatments affect cellular localization, we separated cell extracts into nuclear, cytoplasmic and membrane fractions and probed these with anti-p-SMAD antibodies (Figure 6A). Untreated control cells presented both p-SMAD1/5 forms at the beginning of differentiation (d0) and mainly exhibited nuclear pSMAD1/5^Lo^ during differentiation (d3). By contrast, cells treated with TGFβRII-Fc mainly exhibited cytoplasmic p-SMAD1/5^Hi^ during differentiation (d3). These findings suggest that SMAD1/5^Lo^ can translocate to the nucleus, whereas p-SMAD1/5^Hi^ remains cytoplasmic. We did not observe differences in p-SMAD2/3 localization between control and inhibitor-treated samples but found that p-SMAD2/3 levels increased significantly in inhibitor-treated samples.

To determine how the different p-SMAD forms and treatments affect signaling, we used reporter gene expression assays. Undifferentiated cells exhibited a significant, SMAD1/5/8 dependent BRE reporter signal (Figure 6B) [70]. The signal increased in cells treated with differentiation medium relative to the undifferentiated control at 16- and 48-h. However, cells treated with adipogenesis inhibitors had 4- to 6-fold reduced SMAD1/5/8 signaling relative to untreated controls (Figure 6C). Thus, consistent with our findings of C-terminally phosphorylated p-SMAD1/5^Lo^, SMAD1/5/8 signaling is activated in 3T3-L1 cells and increases further early during differentiation. Anti-adipogenic traps inhibited SMAD1/5/8 signaling, indicating that p-SMAD1/5^Hi^ does not activate BRE dependent transcription. In contrast to SMAD1/5/8, basal SMAD2/3 signaling as measured by the SMAD2/3 responsive SBE reporter [71] was low and minimally induced upon differentiation (Figure 6D). Strikingly, all adipogenesis inhibitors, including Cripto-1-Fc, TGFβRII-Fc and TGF-β1, increased the SMAD2/3 dependent signal approximately 3- to 6-fold relative to untreated controls. Together, these findings indicate that extracellular regulators including Cripto-1-Fc, TGFβRII-Fc, and TGF-β1 arrest adipogenesis by both activating SMAD2/3 signaling and inhibiting SMAD1/5/8 signaling.

### 2.6. Small Molecule Inhibitors Suggest Negative Feedback Loop between SMAD2/3 and SMAD1/5/8 Pathways

Our Western blot and reporter assays showed that both reduced SMAD1/5/8 and increased SMAD2/3 signaling could result in adipogenesis arrest. To define the contribution of each SMAD branch in this process, we investigated the effects of the SMAD2/3 and SMAD1/5/8 activation inhibitors SB-431542 (SB43) and LDN-193189 (LDN) on adipogenesis and signaling (Figure 7) [72,73]. SB43 alone did not have a major effect on adipocyte formation. However, it fully rescued adipogenesis in cells treated with anti-adipogenic traps and ligands (Figure 7A), indicating that SMAD2/3 activation blunts adipogenesis. Strikingly, LDN also inhibited adipogenesis, as evidenced by the complete lack of lipid droplets in treated cells. Together, these results demonstrate that both SMAD2/3 activation and SMAD1/5/8 inhibition led to adipogenesis arrest.

To link SB43 and LDN treatment with SMAD signaling in 3T3-L1 differentiation, we used luciferase reporters. As we found earlier (Figure 6C,D), SMAD2/3 signaling was increased and SMAD1/5/8 signaling was suppressed by TGFβRII-Fc and Cripto-1-Fc (Figure 7B). As expected, SB43 blunted SMAD2/3 signaling in cells treated with inhibitor traps. SB43 also rescued SMAD1/5/8 signaling, indicating that SMAD2/3 activation, SB43, or the SB43 target kinases ALK4, ALK5 or ALK7 may regulate SMAD1/5/8 activities in these cells. In contrast to SB43, LDN suppressed SMAD1/5/8 signaling as expected but had no effect on SMAD2/3 signaling. Strikingly, SB43 did not rescue adipogenesis in LDN treated cells (Appendix A), indicating that SMAD2/3 independent inhibition of SMAD1/5/8 signaling is sufficient to arrest adipogenesis.

To determine at a molecular level how SB43 and LDN elicit their effects on adipogenesis, we probed SMAD activation states using p-SMAD antibodies (Figure 7C). As we showed in Figure 5, control cells only exhibited the p-SMAD1/5^Lo^ species, whereas cells treated with TGFβRII-Fc only exhibited p-SMAD1/5^Hi^. SB43 suppressed formation of p-SMAD1/5^Hi^ in TGFβRII-Fc treated cells, indicating that pathways or targets affected by SB43 may be linked with SMAD1/5/8 hyper-phosphorylation and signaling. Consistent with a critical role for SMAD1/5/8 signaling in adipogenesis, LDN suppressed C-terminal SMAD1/5/8 phosphorylation in cells treated with differentiation medium. Notably, p-SMAD1/5^Hi^ remained detectable in cells co-treated with LDN and TGFβRII-Fc.

Thus, using small molecule inhibitors, we showed that SMAD1/5/8 signaling is essential, whereas SMAD2/3 signaling is detrimental, for 3T3-L1 adipogenesis. In addition, we found that the SMAD2/3 pathway may directly or indirectly regulate SMAD1/5/8 signaling in differentiating 3T3-L1 cells (Figure 8).

## 3. Discussion

Excessive caloric intake relative to energy expenditure stimulates adipose tissue (AT) expansion, a process that comprises adipocyte hypertrophy (increase in adipocyte size) and adipocyte hyperplasia or adipogenesis (increase in adipocyte number) [1]. Adipogenesis is a developmental process triggered by signaling events that direct adipocyte precursors such as mesenchymal stem cells (MSCs) and adipose-derived mesenchymal stromal cells (ADSC) to differentiate into mature adipocytes [74]. Adipogenesis is of particular medical significance as ectopic fat deposition from newly formed adipocytes is strongly correlated with poor metabolic health on other conditions [12,15,75].

TGF-β family pathways feature prominently in adipogenesis [16,17,18]. As such, the family presents potential therapeutic targets that could help regulate formation of new adipocytes. However, TGF-β pathways are multi-functional, redundant and pleiotropic. Understanding how members of the family regulate adipocyte biology and how best to target these pathways in adipogenesis has, therefore, remained a challenge. We addressed this challenge using the 3T3-L1 adipogenesis model, as 3T3-L1 cells exhibit developmental and homeostatic properties of multiple adipocyte lineages [75].

We first compared broadly the effect of various TGF-β family ligands on 3T3-L1 adipogenesis. This comprehensive approach helped us place individual activities within a broader context that allowed us to identify common themes in the mechanisms of adipogenic regulation by the family. Overall, we found that all SMAD2/3 pathway activating ligands can suppress 3T3-L1 adipogenesis either by inhibiting precursor differentiation or by reprograming precursors toward a non-adipogenic lineage. We also found that mature adipocytes became more hypertrophic in their presence, indicating that these ligands could potentially promote lipid storage or alter energy utilization by mature adipocytes. However, a reduced number of mature adipocytes exposed to high levels of nutrients could also explain the observed hypertrophy. By contrast, our results with SMAD1/5/8 pathway activating ligands were more subtle. As the SMAD1/5/8 pathway was activated in the basal state at the beginning of 3T3-L1 differentiation, we speculate that culture conditions provide key BMP growth factors or induce their expression to activate SMAD1/5/8 signaling and, thus, prime undifferentiated 3T3-L1 cells for adipogenic commitment. Further addition of BMPs did not increase the level of 3T3-L1 commitment toward the adipogenic lineage as measured, e.g., by an increased number of lipid droplets relative to the number of cells. However, we observed an overall increase in the number of nuclei, indicating that BMP ligands could promote hyperplastic expansion by increasing adipocyte proliferation at the onset of differentiation [76]. We did not find that BMP ligands promoted adipocyte hypertrophy as measured, e.g., by an increase in lipid droplet size, lipid accumulation, or lipid droplet formation in treated cells. Collectively, these results reveal distinct roles for each SMAD branch in adipogenesis and show that ligands can adopt interchangeable functions in vitro based on their ability to activate a particular SMAD branch. Although it is conceivable that, in vivo, a specific group of ligands primes all precursors for adipogenesis irrespective of their developmental origin, it is also possible that each adipogenic niche may be regulated by its own specific group of ligands.

As one of our goals was to identify adipogenesis inhibitors, we adapted the 3T3-L1 model for in vitro screening of various ligand traps. These traps are engineered by fusing the ligand binding moieties of TGF-β family receptors, co-receptors or antagonists to an antibody Fc domain. They capture distinct groups of ligands in the extracellular space to block ligand–receptor binding and inhibit signaling. We identified three traps that potently suppressed adipogenesis, as 3T3-L1 cells did not form lipid droplets or express adipogenic marker genes in their presence. Intriguingly, TGFβRII-Fc was one of the most potent anti-adipogenic traps. This was unexpected as TGFβRII-Fc is a well-known inhibitor of the SMAD2/3 pathway activating ligands TGF-β1 and TGF-β3, which also suppressed adipogenesis. Other adipogenesis inhibitors identified in this screen were Cripto-1-Fc and mCryptic-Fc, which inhibit Nodal and BMP-4 or Activin B, respectively [57]. Based on the ligand-binding specificities of these three traps, we could not single out one ligand that alone accounted for their effect on adipocyte differentiation. Our results, therefore, suggest that an interplay of multiple ligands may be required to direct 3T3-L1 cells toward adipogenic fates, and that some ligands may act pleiotropically in this context, as they activate their canonical pathways, or as they squelch other ligands and their downstream effectors. Indeed, studies have demonstrated that ligands can act both as signaling activators and antagonists, and that precise ligand combinations may be necessary to provide the signal that leads to the commitment of a cell to a particular fate [56,77,78].

To elucidate the anti-adipogenic mechanism of the different traps and ligands, we investigated their effect on SMAD pathway activation and signal transduction. Using both anti p-SMAD Western blots and reporter gene expression assays, we showed that the SMAD1/5/8 pathway was induced during 3T3-L1 differentiation. All traps and ligands that blunted adipogenesis suppressed SMAD1/5/8 signaling as measured by reporter gene expression. Consistent with these findings, the SMAD1/5/8 signaling inhibitor LDN also suppressed 3T3-L1 adipogenesis. However, using anti p-SMAD Western blots, we discovered that, unlike LDN, traps and ligands did not inhibit SMAD1/5/8 signaling simply by blocking SMAD1/5/8 C-terminal phosphorylation. Instead, we observed the near complete conversion of p-SMAD1/5/8 into a higher molecular weight p-SMAD1/5^Hi^ form that could be restored to its original electrophoretic mobility with alkaline phosphatase treatment. As TGFβRII-Fc treated cells predominantly exhibited p-SMAD1/5^Hi^ and this form was mainly found in the cytoplasmic fraction, we speculate that p-SMAD1/5^Hi^ is hyper-phosphorylated SMAD1/5/8 that fails to translocate to the nucleus and, thus, may have reduced signaling activity. In support of this model, several studies have shown that SMADs harbor a number of phosphorylation sites, which are associated with altered cellular localization and signaling [66,77,78,79,80]. However, we cannot rule out other SMAD1/5/8 post-translational modifications or cross-reactivity of the p-SMAD1/5/8 antibody with p-SMAD2/3. Nevertheless, we show that SMAD1/5/8 signaling is vital for adipogenesis. Our data further indicate that SMAD1/5/8 hyper-phosphorylation may represent a distinct mechanism of SMAD1/5/8 pathway regulation in these cells.

In contrast to SMAD1/5/8, SMAD2/3 signaling was suppressed at all times during 3T3-L1 differentiation. As all anti-adipogenic traps and ligands, including TGFβRII-Fc, activated SMAD2/3 signaling, we propose that SMAD2/3 activation is deleterious for adipogenesis. Consistent with this idea, the SMAD2/3 activation inhibitor SB43 blunted the anti-adipogenic effect of inhibitory traps and ligands. Strikingly, SB43 also restored SMAD1/5/8 signaling and suppressed formation of the p-SMAD1/5^Hi^ species in cells treated with TGFβRII-Fc. These results, therefore, indicate that SMAD2/3 activation and/or the SB43 target kinases ALK4, ALK5 or ALK7 may be linked with SMAD1/5/8 hyper-phosphorylation and signaling inhibition. Indeed, TGF-β1 has been shown to inhibit BMP7 mediated transcriptional responses [81,82]. Alternatively, SB43 target kinases could activate intracellular kinases or other pathways that are linked with SMAD hyper-phosphorylation [83], and we observed increased ERK kinase phosphorylation in cells treated with TGFβRII-Fc. However, ERK inhibitors failed to restore adipogenesis or prevent SMAD1/5/8 hyper-phosphorylation, suggesting a limited role of ERK kinase in this context. Thus, we speculate that SMAD1/5/8 signaling and/or hyper-phosphorylation could be mediated via a regulatory feedback loop that links SMAD2/3 activation with SMAD1/5/8 inhibition. Notably, inhibition of SMAD1/5/8 with LDN did not alter SMAD2/3 activities, and SB43 did not rescue adipogenesis in cells treated with LDN, indicating that the feedback loop may only work in one direction and that inhibition of SMAD1/5/8 is necessary to block 3T3-L1 adipogenesis.

## 4. Materials and Methods

**Ligands.** Human Activin A, Activin B, GDF-8, GDF-11, TGF-β1, BMP-2, BMP-4, BMP-6, BMP-9, and BMP-10 were obtained from R&D Systems, PROMOCELL or produced in-house. Activity was verified by Surface Plasmon Resonance and reporter gene assays.

**Fc-Fusion Proteins.** Synthetic genes of human ActRIIA, ActRIIB, ALK2, ALK3, ALK4, Cripto-1, and Cerberus, as well as mouse Cryptic were obtained from GeneArt. Human BMPRII and TGFβRII were PCR amplified from cDNA (Open Biosystems). Extracellular domains were fused to IgG1-Fc by PCR. Fc-fusion proteins were expressed using Chinese hamster ovary (CHO) cells, captured from condition medium using protein A affinity chromatography, eluted with 100 mM glycine, pH 3.0, and directly neutralized by adding 2% *v*/*v* 2 M Tris/HCl, pH 9.0. Purified proteins were either dialyzed directly into phosphate-buffered saline, pH 7.5, and stored at −80 °C, or further purified by size exclusion chromatography in phosphate-buffered saline, pH 7.5, and stored at −80 °C. Purity was determined by SDS-PAGE and activity was verified by Surface Plasmon Resonance.

**Small Molecule Inhibitors.** SB-431542 (SB43, 10 μM), LDN-193189 (LDN, 1 μM), PD98059 (PD98, 1 μM), PD0325901 (PD03, 1 μM), and the selective ERK inhibitor FR180204 (FR18, 10 μM) were purchased from Biovision. Samples were reconstituted in DMSO according to the manufactures’ instructions.

**3T3-L1 Differentiation Assay.** Murine 3T3-L1 pre-adipocytes were purchased from ZenBio. Cryopreserved 3T3-L1 cells were thawed and seeded at approximately 10,000 cells/cm^2^ in Preadipocyte Medium (PM: DMEM, high glucose, HEPES pH 7.4, 10% Bovine Calf Serum (BCS), and Penicillin + Streptomycin (PS)). Cells were maintained at 37 °C in a humidified incubator with 5% CO_2_ until reaching 100% confluence (in about 4 days). During this time, media was replaced every other day. Two days after reaching confluence, Preadipocyte Medium (PM) was replaced with an appropriate volume of Differentiation Medium (DM: DMEM, high glucose, sodium pyruvate, HEPES pH 7.4, 10% Fetal Bovine Serum (FBS), 33 µM Biotin, 10 µg/mL Human insulin, 1 µM Dexamethasone, 0.5 mM 3-Isobutyl-1-methylxanthine (IBMX)) and incubated for 3 days. Differentiation Medium was then replaced with Adipocyte Maintenance Medium (MM: DMEM high glucose, sodium pyruvate, HEPES pH 7.4, 10% FBS, 33 µM Biotin, 10 µg/mL Human insulin,). Cells were maintained up to 10 days post differentiation with medium exchange every other day.

**Treatments.** 3T3-L1 cells were treated beginning at different stages of differentiation. Treatments were generally maintained until harvest. Briefly, confluent 3T3-L1cells were grown in PM, differentiated 3 days in DM and maintained up to 5 days in MM. Either 1 nM ligand (except TGF-β1, which is toxic at 1 nM as determined in a dose response assay, used at 0.1 nM) or 300 nM traps were added at day 0 (beginning of DM treatment), day 3 (end of DM treatment) or day 5 (after 2 days in MM). Cells were kept under treatment until the end of the experiment at day 8.

**Immunofluorescence.** A total of 10,000 3T3-L1 cells/cm^2^ were plated in a 96-well plate in preadipocyte media. At day 0, cells were treated with differentiation medium containing test articles or small molecule inhibitors. Treatments were continued in the appropriate medium containing test articles as specified in the ’3T3-L1 Differentiation assay’ section until day 10. At day 10, cells were washed twice with PBS and fixed with 10% formalin for 30 min at RT. Cells were then washed twice with PBS, followed by staining with 0.01% saponin, 1 µg/mL Nile Red and 1 µg/mL DAPI in PBS for 15 min at RT. After staining, cells were washed 3 times with PBS. Images were taken with Olympus Fluoview FC1000 confocal laser scanning microscope. In the figures, green represents Nile Red staining, purple represents DAPI staining. For quantitative Nile Red and DAPI fluorescence measurements, fluorescence was measured before and after staining according to published protocol [84]. Published data were obtained in duplicate. Assays were repeated multiple times.

**Image Analysis.** 3T3-L1 cells were treated in quadruplicates in 96 well plate. Multiple images were taken from each well. Number of lipid droplets, number of nuclei, and mean lipid droplet intensity were calculated using ImageJ software from two biological replicates.

**qRT-PCR.** Genomic DNA was isolated from 3T3-L1 cells and standard curves were created using a 1/3-fold dilution series with the highest concentration yielding qRT-PCR amplification at around cycle 20 for the majority of primer pairs. DNA and cDNA was quantified on a Lightcycler 480 (Roche) as described [58]. Primers were designed using the program PCRTiler [85]. A Tm-curve was performed as a quality control for each primer pair at the end of each qRT-PCR run to verify that only a single amplicon was produced by each primer pair and that no primer dimers were formed. Published data were obtained from two biological replicates.

**Immunoblots.** Approximately 19,000 3T3-L1 cells were plated in 24-well plates and grown to confluence in preadipocyte medium. Cells were then switched into differentiation and maintenance medium as required. The medium contained test articles, including ligands (0.1–1 nM), Fc-fusion traps constructs (−300 nM) and/or small molecule inhibitors (1–10 µM). After 3 to 10 days of cell growth at 37 °C, protein lysate was prepared by using ice-cold RIPA lysis buffer (150 mM NaCl, 1% Nonidet P-40, 0.1% SDS, 0.5% sodium deoxycholate, 50 mM Tris, pH 8.0, 1X “Protease Arrest” and 2X “Phosphatase Arrest” (G-Biosciences). Cell lysates were stored at −80 °C. Total protein concentration was determined by Bradford. For Western blot, equal amounts of protein (typically 10 μg) were separated under reducing conditions on 10 or 12% TGX-polyacrylamide gels (Bio-Rad) and transferred to Hybond-P membrane (GE Healthcare). Membranes were blocked with Superblock (Thermofisher) and incubated with primary antibodies from Cell Signaling at a 1:1000 dilution (e.g., anti-phospho-SMAD2/3 (138D4), anti-phospho-SMAD1/5/8 (41D10)), or 1:5000 (e.g., anti-β-actin (8H10D10)), followed by incubation with horseradish peroxidase-conjugated secondary antibody at dilutions of 1:10,000 (Actin) and 1:2000 (SMADs) dilution (7074). Western Bright ECL HRP substrate was used for detection (Advansta). Blots were visualized using autoradiography film. A list of antibodies can be found in Appendix A. For IP, 250 µg protein in total cell lysate was immunoprecipitated with 0.175 µg anti-p-SMAD1/5/8 antibody. Eluted samples were analyzed by Western Blot using as secondary antibody both conformation specific and standard anti-Rabbit Fc antibodies.

**Reporter Gene Expression.** 3T3-L1 cells in preadipocyte medium were seeded in each well of a 96-well plate and grown overnight. For transfection, solutions containing 0.2 µL/well Lipofectamine 3000, 0.2 µL/well P3000, 1 ng/well pGL4.74 plasmid (Luc2P/hRluc/TK, control luciferase reporter plasmid, Promega), and 100 ng/well of SMAD2/3 responsive reporter plasmid pGL4.48 (luc2P/SBE) or SMAD1/5/8 responsive reporter plasmid pGL3 (luc2P/BRE) in Opti-MEM were incubated at room temperature for 15 min. Transfection mixture was added to preadipocyte or differentiation medium containing test articles. Cells were then incubated for 16 h at 37 °C. Luciferase activity was detected using a homemade dual-glow luciferase assay [86]. Luminescence was determined using a FLUOstar Omega plate reader. Relative luciferase units were calculated by dividing firefly luciferase units with *Renilla* luciferase units (RLU). Data are expressed as mean of four independent measurements. Error bars correspond to S.E. of four independent measurements.

**Alkaline Phosphatase Treatment.** A total of 440,000 3T3-L1 cells were plated in a 100 mm dish and grown to confluence in preadipocyte medium. Cells were then switched into differentiation medium containing test articles at day 0. At day 3, cells were scraped in 50 mM Tris pH 8.0, 100 mM NaCl, 10 mM MgCl_2_, 1 mM DTT buffer and sonicated. Lysate concentration was determined by Bradford. A quantity of 10 µg of lysate was treated with 0.5 µL alkaline phosphatase (Sigma Aldrich, P0114) and incubated at 37 °C for 2 h. After incubation, samples were run on 10% TGX-polyacrylamide gels (Bio-Rad) and blotted with different antibodies.

**Cellular Localization.** Subcellular fractionation of 3T3-L1 cells was carried out according to published methods (https://bio-protocol.org/e754, 5 May 2013). Briefly, 440,000 3T3-L1 cells were plated in a 100 mm dish and grown to confluence in preadipocyte medium. Two days after plating, cells were switched into preadipocyte medium containing test articles. At day 0, cells were switched into differentiation medium containing test articles. Cells were harvested either at day 0 or day 3 in subcellular fractionation buffer (250 mM sucrose, 20 mM HEPES pH 7.4, 10 mM KCl, 1.5 mm MgCl_2_, 1 mM EDTA,1 mM EGTA, 1 mM DTT, 1X “Protease Arrest” and 2X “Phosphatase Arrest” (G-biosciences). Cytoplasmic, nuclear and membrane fractions were separated as described (https://bio-protocol.org/e754, 5 May 2013). Equal total protein amounts of cytoplasmic, nuclear and membrane fractions were separated on 10% TGX gels (Bio-Rad) and blotted with different antibodies.

**Proliferation and Apoptosis Assays.** For the proliferation and apoptosis assay, 3T3-L1 cells were plated in 96 well plates at a cell density of 10,000 cells/cm^2^ in preadipocyte media. At day 0, cells were treated with differentiation medium containing either TGF-β1 or TGβRII-Fc or BMP-6. Measurements were taken at different time points (0 h, 24 h, and 72 h). Experiments were performed in triplicate and repeated with two or three separately initiated cultures. For cell viability, cells were trypsinized and mixed with Trypan Blue. Cell counts were obtained using a Bio-Rad TC10 cell counter. For the MTT proliferation assay, cells were incubated with MTT solution to a final concentration of 0.5 mg/mL. Unreacted dye was removed after 2-h incubation and the insoluble formazan crystals were dissolved in DMSO. The absorbance was read at 560 nm using FLUOstar plate reader. For the BrdU proliferation assay, cells were cultured with treatment conditions for indicated time period and pulsed with BrdU for 4 h before the end of each experimental period. BrdU incorporation was assayed using a commercial kit (Cell signaling). For apoptosis, Caspase3/7 activity was measured using the Caspase-Glo kit (Promega). The luminescence was measured using FLUOstar plate reader.

## 5. Conclusions

We showed that SMAD1/5/8 pathways are activated while SMAD2/3 pathways are suppressed in differentiating 3T3-L1 cells. We identified several ligand traps that prevent 3T3-L1 adipogenesis via a conserved regulatory mechanism that involves both activation of SMAD2/3 and inhibition SMAD1/5/8 signaling. Overall, these findings offer key insights into the complexity of TGF-β family action and regulation in adipogenesis. Importantly, we have identified several ligand traps with anti-adipogenic activity in vitro, including TGFβRII-Fc, Cripto-1-Fc and mCryptic-Fc, which could help control hyperplastic AT expansion in vivo.

## Figures and Tables

**Figure 1 ijms-22-08472-f001:**
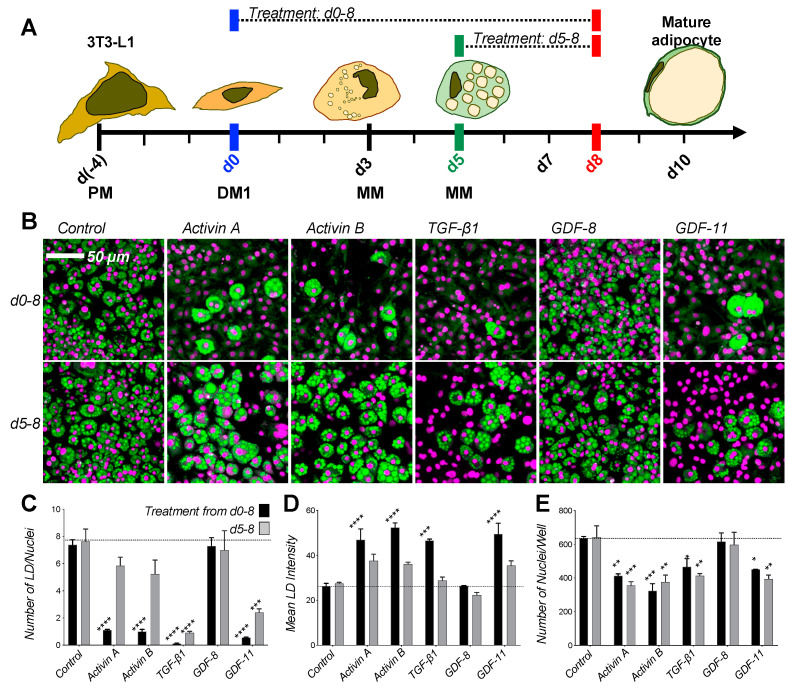
TGF-βs and activins inhibit differentiation of 3T3-L1 cells into adipocytes. (**A**) Schematic of 3T3-L1 assay. Cells are grown 4 days in Preadipocyte Medium (PM) (from day -4 to 0), differentiated for 3 days using Differentiation Media, and maintained up to 7 days in Maintenance Medium (MM from day 3 to 10). Cells are treated with ligands or traps either from day 0 (blue) or from day 5 (green) until harvest at day 8 (red). (**B**) 3T3-L1 cells were grown with 1 nM ligand as noted (except TGF-β1 at 0.1 nM) or vehicle control (PBS) from day 0 (top panel) or day 5 (bottom panel) of differentiation until day 8. Cells were fixed at day 8 and stained for lipids using Nile red (green), nuclei were counter-stained with DAPI (magenta). Cells treated with this group of ligands mostly showed significantly reduced lipid droplet formation. (**C–E**) Quantitative analysis of 3T3-L1 samples treated with ligands from day 0 (black) or day 5 (grey) of differentiation. Images were analyzed with ImageJ and data were evaluated using Prism 9. Statistical significance from two biological replicates was calculated by two-way ANOVA and Dunnett’s multiple comparisons test (* *p* < 0.05; ** *p* < 0.01; *** *p* < 0.001; **** *p* < 0.0001). (**C**) Total number of lipid droplets (LD) per well. (**D**) Mean lipid droplet intensity. (**E**) Number of nuclei per well.

**Figure 2 ijms-22-08472-f002:**
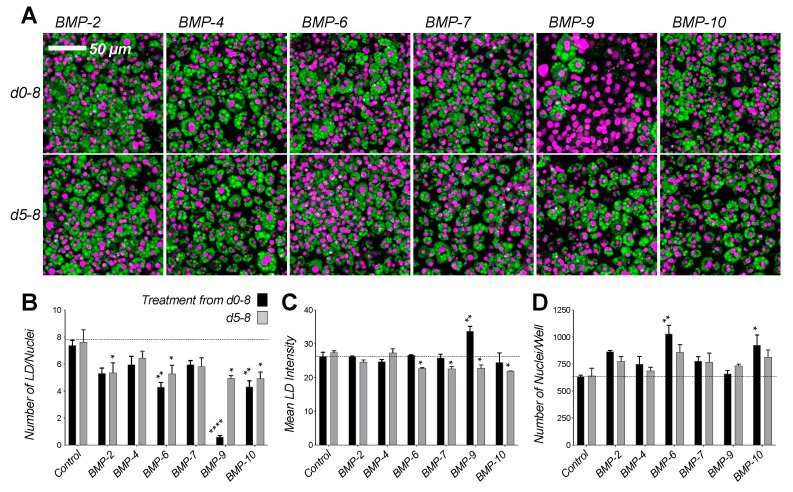
BMP-ligands induce 3T3-L1 proliferation. (**A**) 3T3-L1 cells were grown with 1 nM ligand as noted or vehicle control (PBS) from day 0 (top panel) or day 5 (bottom panel) of differentiation until day 8. Vehicle control is shown in Figure 1B. Cells were fixed at day 8 and stained for lipids using Nile red (green), nuclei were counter-stained with DAPI (magenta). Cells treated with various BMPs generally show increased number of nuclei and DAPI fluorescence. (**B**–**D**) Quantitative analysis of 3T3-L1 samples treated with ligands from day 0 (black) or day 5 (grey) of differentiation. Confocal images were analyzed using ImageJ and data were evaluated using Prism 9. Statistical significance from two biological replicates was calculated by two-way ANOVA and Dunnett’s multiple comparisons test (* *p* < 0.05; ** *p* < 0.01; **** *p* < 0.0001). (**B**) Total number of lipid droplets (LD) per well. (**C**) Mean lipid droplet intensity. (**D**) Number of nuclei per well.

**Figure 3 ijms-22-08472-f003:**
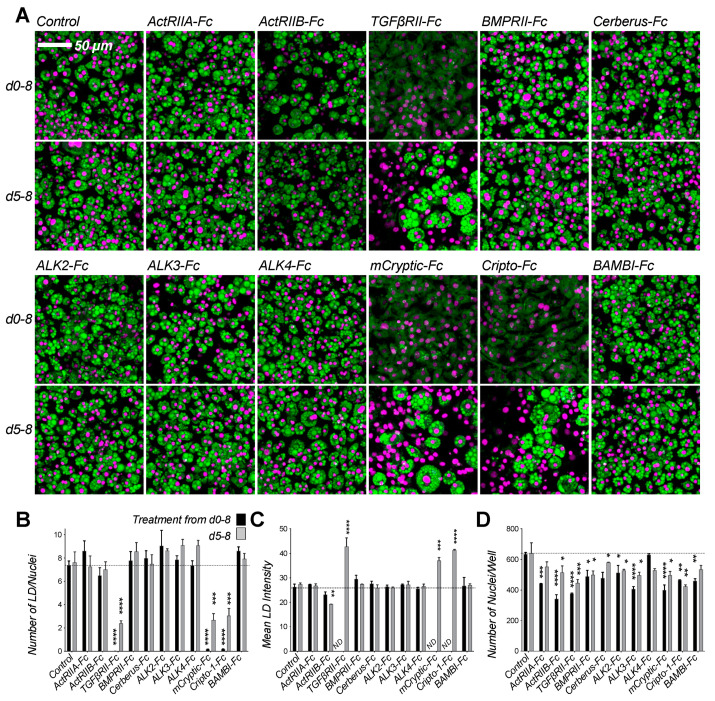
Differentiation assay screen identifies traps with anti-adipogenic activity. (**A**) 3T3-L1 cells were grown in the presence of 300 nM Fc-fusion traps or vehicle control (PBS) as noted. Cells were treated from day 0 (top panel) or day 5 (bottom panel) of differentiation until day 8. Cells were fixed at day 8 and stained for lipids using Nile red (green), nuclei were counter-stained with DAPI (magenta). Cells treated with TGFβRII-Fc, mCryptic-Fc, or Cripto-1-Fc show significantly reduced lipid droplet formation. Each trap captures a unique group of ligands (ST1). Quantitative analysis of 3T3-L1 samples treated with ligands beginning at day 0 (black) or day 5 (grey) of differentiation. Confocal images were analyzed using ImageJ and Prism 9. Statistical significance from two biological replicates was calculated by two-way ANOVA and Dunnett’s multiple comparisons test (* *p* < 0.05; ** *p* < 0.01; *** *p* < 0.001; **** *p* < 0.0001). (**B**) Total number of lipid droplets (LD) per well. (**C**) Mean lipid droplet intensity. (**D**) Number of nuclei per well.

**Figure 4 ijms-22-08472-f004:**
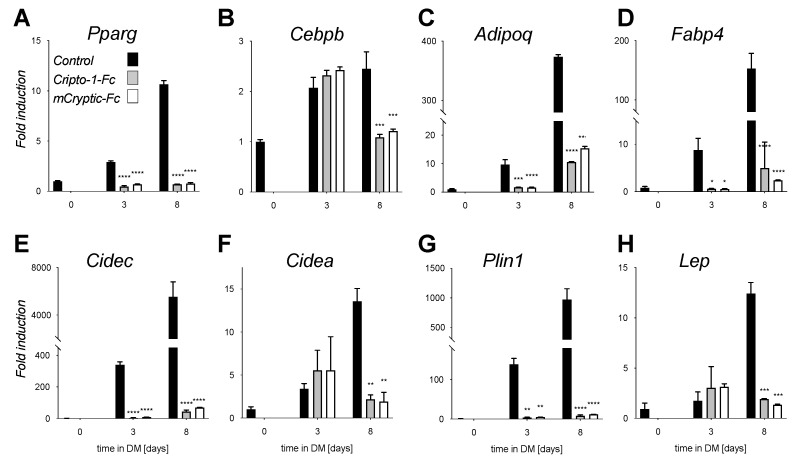
Inhibitory traps suppress expression of adipocyte marker genes. (**A**–**H**) Induction of adipocyte marker gene expression was analyzed by qRT- PCR on days 0, 3 and 8 of differentiation in control, mCryptic-Fc, and Cripto-1-Fc treated cells (black, grey and white bars, respectively). RNA isolation and qRT-PCR analysis were performed as described [42]. Data was normalized to Rpl4 mRNA and is shown as fold induction relative to day 0 levels. Statistical significance from two biological replicates was determined by two-way ANOVA and Sidaki’s post-hoc tests (* *p* < 0.05; ** *p* < 0.01; *** *p* < 0.001; **** *p* < 0.0001) using Prism 9. Expression of adipogenic transcription factors (**A**) Pparg, and (**B**) Cebpb. Expression of adipocyte marker genes (**C**) Adipoq, (**D**) Fabp4, and (**E**) Lep. Expression of genes associated with lipid droplet formation (**F**) Cidec, (**G**) Cidea, and (**H**) Plin1.

**Figure 5 ijms-22-08472-f005:**
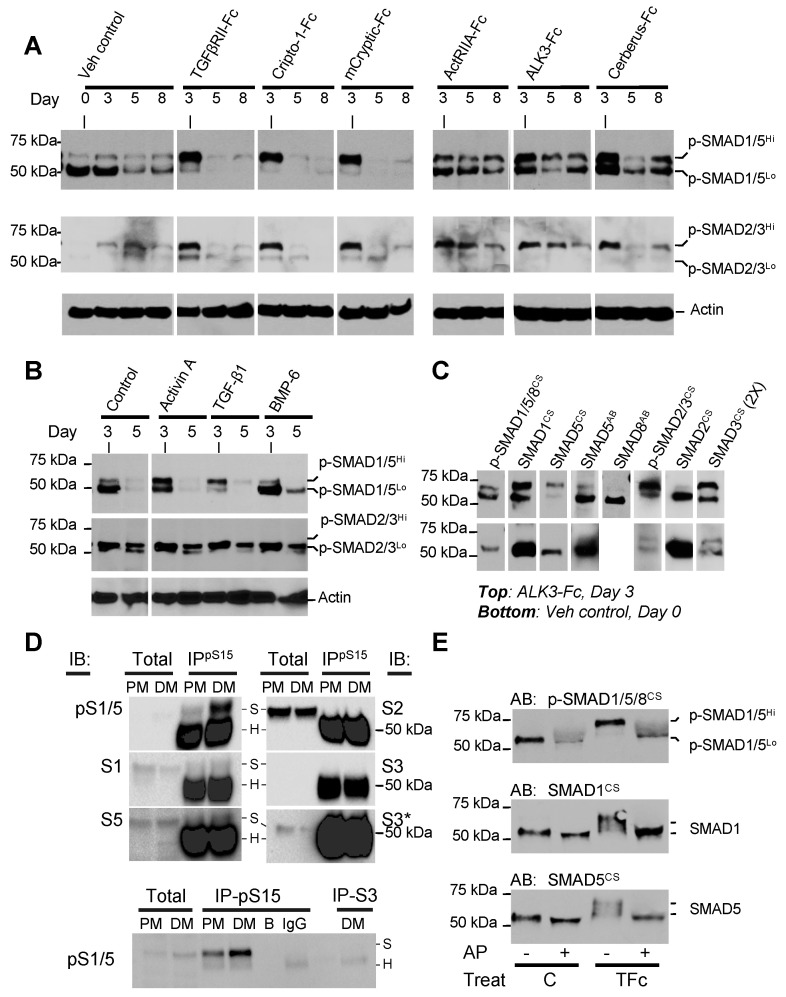
Altered SMAD activities and electrophoretic mobility shift in p-SMAD1/5 associate with anti-adipogenic mechanism. (**A**) Anti-p-SMAD Western blots of whole cell lysate show C-terminal Serine phosphorylation of SMADs (top panel: p-SMAD1/5/8, middle panel: p-SMAD2/3) in samples treated with different ligand traps or vehicle (PBS). Treatments were started at day 0 of differentiation and samples were collected at days 3, 5, or 8 of differentiation as noted. Two p-SMAD1/5/8 forms were detected and are labeled p-SMAD1/5^Lo^ and p-SMAD1/5^Hi^, reflecting differences in electrophoretic mobility. Similarly, p-SMAD2/3 also appears as two bands. Actin Western blots were used as sample loading controls. Blots were loaded with 10 µg protein per lane. Quantification is shown in Appendix A. Total SMAD loading controls are shown in Appendix A. (**B**) Anti-p-SMAD Western blots of whole cell lysate show C-terminal Serine phosphorylation of SMADs (top panel: p-SMAD1/5/8, middle panel: p-SMAD2/3) in samples treated with different ligands and Fc control (ALK3-Fc). Treatments were started at day 0 of differentiation. Samples were loaded at a 3-fold higher amount than in panel A to make up lower detection or lower abundance issues. Samples were collected at days 3 or 5 of differentiation. Actin was used as loading control. Blots for Actin and p-SMAD1/5/8 have 10 µg protein per lane. The p-SMAD2/3 blot has 30 µg protein per lane. Quantification is shown in Appendix A. Total SMAD loading controls are shown in Appendix A. Appendix A shows receptor levels. (**C**) Anti-SMAD Western blots of whole cell lysate show overall SMAD levels in ALK3-Fc treated samples (top panel, sample collected at day 3) and vehicle control samples (PBS, bottom panel, sample collected at day 0). ALK3-Fc was used as positive control as both p-SMAD1/5^Lo^ and p-SMAD1/5^Hi^ forms were prominently visible with this treatment. Antibodies used are noted above the blots. CS denotes an antibody from Cell Signaling Technologies, AB denotes an antibody from Abcam. p-SMAD antibodies recognize conserved, C-terminally phosphorylated Serine residues in activated SMADs. All other SMAD antibodies were raised against unique sequences within each SMAD protein. Blots were loaded with 10 µg protein per lane. (**D**) Anti-p-SMAD1/5 immunoprecipitation. A quantity of 250 µg of whole cell lysate collected before differentiation (PM) and at day 3 of differentiation (DM) was IP’d using an anti-p-SMAD1/5 antibody. Each Immunoblot (IB) contained the total eluted fraction. Samples were blotted with anti-pSMAD1/5 (pS1/5), SMAD1 (S1), SMAD2 (S2), SMAD3 (S3), and SMAD5 (S5) as primary antibody and a standard secondary antibody (top panel), as well as a conformation specific antibody (bottom panel). The S3 IB is also shown at a higher exposure (S3*) to better visualize SMAD3. S denotes bands corresponding to the respective SMAD, H denotes bands corresponding to the antibody heavy chain. p-SMAD1/5, but not the heavy chain, is detected using the conformation specific antibody (bottom panel). The bottom panel also shows bead control (B) and IgG control (IgG). The bottom panel also shows IP with a SMAD3 specific antibody (S3) and IB’d with anti-pSMAD1/5. (**E**) Phosphorylation analysis by Alkaline phosphatase digestion. Western blots of whole cell lysate show Vehicle control (C) and TGFβRII-Fc (TFc) treated samples subjected to Alkaline phosphatase (AP) dephosphorylation (+). Antibodies against C-terminally phosphorylated SMAD1/5/8 (p-SMAD1/5/8), SMAD1 and SMAD5 were used as noted. The p-SMAD1/5^Hi^ form found in TGFβRII-Fc treated samples reduces to the p-SMAD1/5^Lo^ form upon AP digestion. A total of 10 µg of protein per lane were used. WB quantifications are shown in Appendix A.

**Figure 6 ijms-22-08472-f006:**
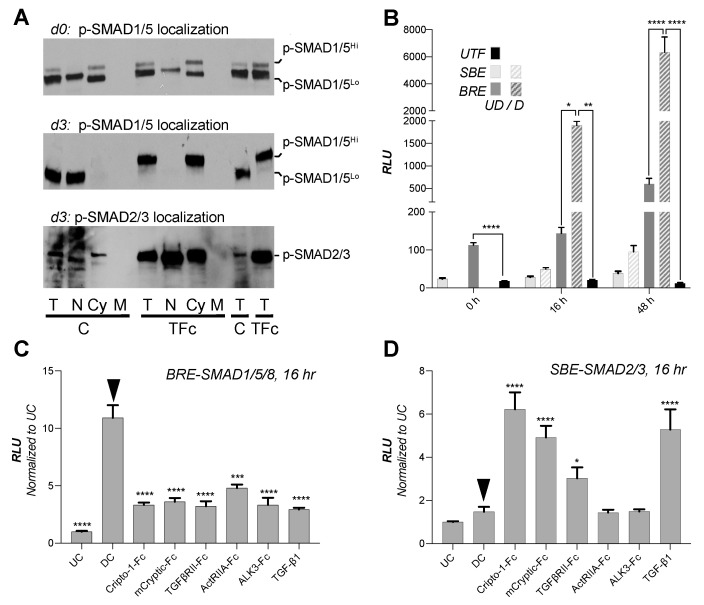
Electrophoretic mobility shift is associated with p-SMAD1/5 hyper-phosphorylation and leads to reduced p-SMAD1/5 nuclear translocation. (**A**) SMAD localization by cellular fractionation. Total cell lysate (T), nuclear (N), cytoplasmic (Cy), and membrane (M) fractions of vehicle control (C) and TGFβRII-Fc (TFc) treated samples were analyzed by Western blot after cellular fractionation using p-SMAD1/5/8 and p-SMAD2/3 antibodies. The top panel and the middle and bottom panels show samples with treatment beginning at d-2 and d0 and collected at d0 and d3 of differentiation, respectively. p-SMAD1/5^Hi^ is found in the cytoplasmic fraction in all TGFβRII-Fc treated samples. Only p-SMAD1/5^Lo^ is found in the nuclear fraction. A total of 10 µg of protein, as determined by per lane, were loaded. WB quantifications are shown in Appendix A. Total SMAD loading controls, and tubulin localization controls are shown in Appendix A. (**B**) Dual luciferase reporter assay in 3T3-L1 cells shows basal SMAD1/5/8 and SMAD2/3 signaling as noted at 0 (light grey) and 16 (dark grey) hours of differentiation. Samples represented by solid bars were undifferentiated (UD), samples represented by stripped bars were treated with differentiation reagents (D). Firefly luciferase reporter lacking control (untransfected, UTF) is shown as black bar. Data were analyzed using Prism 9 and statistical significance from four biological replicates was determined by one-way ANOVA and Fisher’s LSD tests (* *p* < 0.05; ** *p* < 0.01; *** *p* < 0.001; **** *p* < 0.0001). Measurements were taken at 0, 12 and 48 h post differentiation as noted. (**C**) Dual luciferase reporter assay in 3T3-L1 cells shows SMAD1/5/8 signaling at 16 h of differentiation. Samples were treated with different ligands or traps as noted. Both reporter plasmid transfection and treatment started at 0 h of differentiation. The SMAD1/5/8 dependent firefly luciferase signal was normalized against Renilla luciferase control. Data are shown as fold induction relative to undifferentiated control (UC). Data were analyzed using Prism 9. Statistical significance from four biological replicates was determined by one-way ANOVA and Fisher’s LSD tests by comparing treatments against differentiated control (DC, black arrow; * *p* < 0.05; ** *p* < 0.01; *** *p* < 0.001; **** *p* < 0.0001). (**D**) Dual luciferase reporter assay in 3T3-L1 cells shows SMAD2/3 signaling at 16 h of differentiation. Samples were treated with different ligands or traps as noted. Both reporter plasmid transfection and treatment started at 0 h of differentiation. The SMAD2/3 mediated firefly luciferase signal was normalized against the Renilla luciferase internal control. Data are shown as fold induction relative to undifferentiated control (UC). Data were analyzed using Prism 9. Statistical significance from four biological replicates was determined by one-way ANOVA and Fisher’s LSD tests by comparing treatments against differentiated control (DC, black arrow; * *p* < 0.05; ** *p* < 0.01; *** *p* < 0.001; **** *p* < 0.0001).

**Figure 7 ijms-22-08472-f007:**
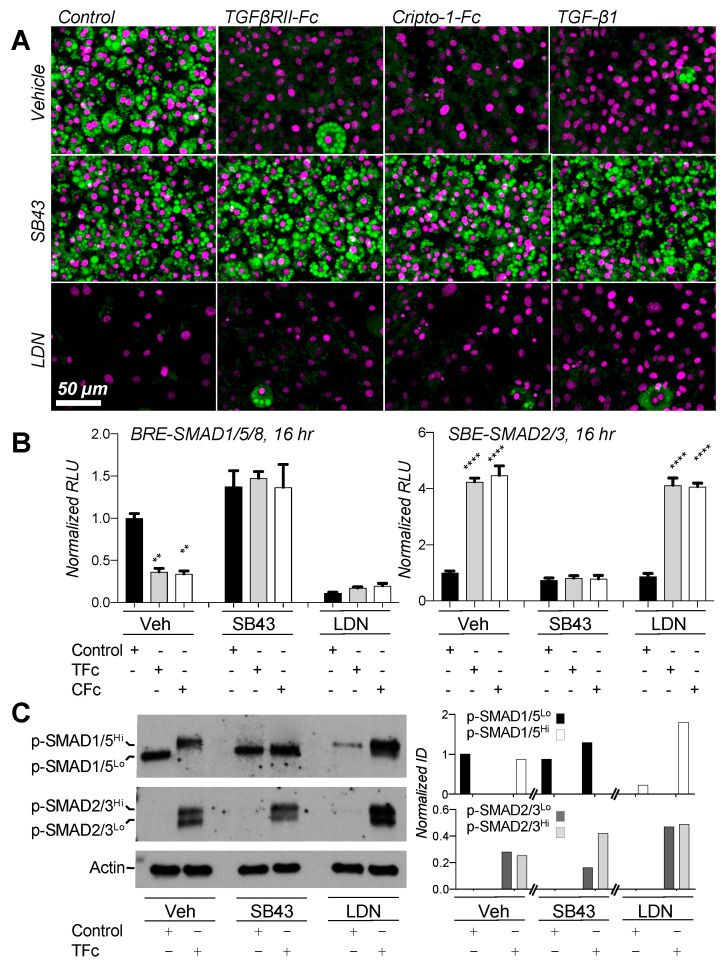
Effect of small molecule inhibitors on 3T3-L1 differentiation and signaling. (**A**) 3T3-L1 cells were grown in the presence of 300 nM Fc-fusion traps or vehicle control (PBS) and the small molecule inhibitors SB-43142 (SB43, 10 μM) or LDN 193189 (LDN, 1 μM) as noted. Cells were treated from day 0 of differentiation until collected. Cells were collected at day 8, fixed and stained for lipids using Nile red (green). Nuclei were counter-stained with DAPI (magenta). SB43 rescues cells treated with adipogenesis inhibitors, while LDN suppresses adipogenesis as indicated by the respective presence and absence of lipid droplets (green). (**B**) Dual luciferase reporter assay shows BRE-SMAD1/5/8 (left panel) and SBE-SMAD2/3 (right panel) mediated signaling in 3T3-L1 cells at 16 h of differentiation. Black bars correspond to vehicle, light grey bars to TGFβRII-Fc, white bars to Cripto-1-Fc treatment. Groups are separated by secondary treatment as noted, including Veh (DMSO), SB43 and LDN. SMAD2/3 or SMAD1/5/8 dependent firefly luciferase signals were normalized against their respective Renilla luciferase internal controls. All treatments were then normalized to Vehicle controls. Controls in vehicle group were used as reference for analysis in Prism 9. Statistical significance from four biological replicates was determined by two-way ANOVA and Tukey’s multiple comparison test (significance relative to control samples (black) is as follows: ** *p* < 0.01; **** *p* < 0.0001). SB43 induces BRE-SMAD1/5/8 luciferase activity. LDN does not affect SBE-SMAD2/3 luciferase activity. (**C**) Anti-p-SMAD Western blots of 3T3-L1 cells treated with small molecule inhibitors. The top panel shows p-SMAD1/5/8, the middle panel p-SMAD2/3, the bottom panel actin loading controls. Control and TGFβRII-Fc treated cells were subjected to additional Veh (DMSO), SB43 and LDN treatment from day 0 of differentiation and samples were collected at day 3. SB43 prevents SMAD1/5/8 hyper-phosphorylated in TGFβRII-Fc treated cells, as evidenced by the absence of pSMAD1/5^Hi^. WB quantifications are shown in Appendix A.

**Figure 8 ijms-22-08472-f008:**
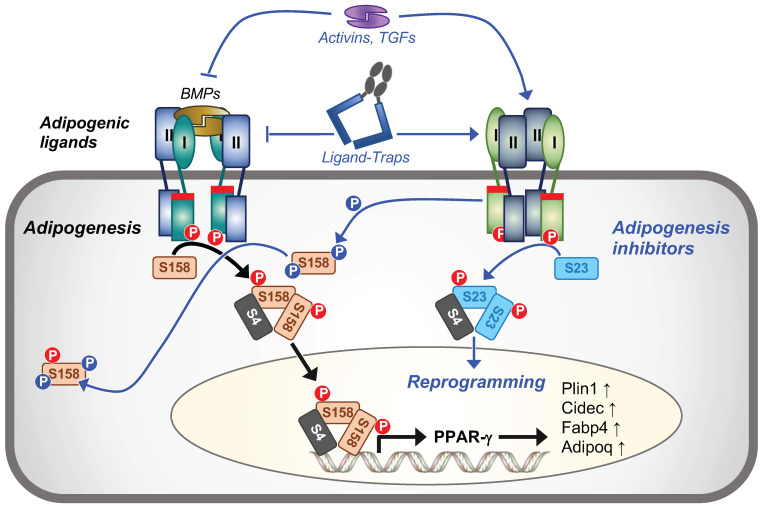
SMAD signaling and regulation in adipogenesis. Black arrows represent the proposed pathway of BMP/TGF-β signaling that leads to adipogenesis. BMPs and the type I receptor kinases, including ALK2, and ALK3 (Appendix A) activate SMAD1/5/8 signaling (orange rectangles labeled S158) by C-terminal phosphorylation (red circled P). Activated SMAD1/5/8 form a trimeric complex with SMAD4 (grey rectangles labeled S4), which translocates to the nucleus to directly or indirectly regulate expression of adipogenic genes, including Pparg, Plin1, Cidec, Fabp4 and Adipoq. SMAD2/3 signaling (light blue rectangles labeled S23) is inactive in adipogenesis. Blue arrows represent the proposed pathway of BMP/TGF-β signaling in adipogenesis arrest. Differentiating 3T3-L1 cells treated with anti-adipogenic traps and ligands show both SMAD2/3 activation and SMAD1/5/8 inhibition. SMAD2/3 activating kinases or SMAD2/3 regulated genes could negatively regulate SMAD1/5/8 signaling by inducing its hyperphosphorylation (blue circled P), which may lead to reduced SMAD1/5/8 nuclear translocation and transcriptional activity.

## Data Availability

Data is contained within the article or Appendix A.

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
