# Peer review of "Smad2/3 Activation Regulates Smad1/5/8 Signaling via a Negative Feedback Loop to Inhibit 3T3-L1 Adipogenesis"

_ijms, 2021, doi:10.3390/ijms22168472_

Round 1

Reviewer 1 Report

The article entitled ' Smad2/3 activation regulates Smad1/5/8 signalling via a negative feedback loop to inhibit 3T3-L1 adipogenesis’ by Aykul et al is very interesting and provides compelling evidence of the role of BMP signalling in adipogenesis.  The authors needs to address the below comments.

 Comments

  1. The article suggested that LDN-193189 inhibitor suppresses adipogenesis, by inhibiting the kinase activity of the BMP receptors, however, the LDN-193189 inhibitor also inhibits p38 signalling orchestrated by TGFbeta via the non-smad signalling cascade (TAK1-TRAF6-P38 cascade). Can the authors provide evidence that this is solely dependent on BMP signalling?
  2. In Fig 5 A, B, C, D, E. The WB data needs total SMAD2/3 blots and the respective receptors (TBRII, ActRIIA etc) immunoblots. For D and E, total SMAD 1/5/8. Densitometry of WB data is required to interpret the data.
  3. The Mol size of the proteins needs to mentioned in the immunoblots.
  4. Fig 6A, total SMAD 1/5 , SMAD 2/3 is missing.
  5. How do we differentiate localization of proteins in nuclear or cytoplasmic compartments? Can the authors probe the same samples for laminin A, a marker for nuclear fraction and tubulin, a marker for cytoplasmic fraction?
  6. Fig 6A, labelling of the samples in each well is misaligned, Please format it.
  7. In Fig 6B, The Y axis information is missing (the time points correspond to?)
  8. Fig 7, total SMAD 1/5 , SMAD 2/3 is missing.
  9. In Fig 7, In the legend text, the conc of the inhibitor used is described in M. please clarify
  10. Number of experiments performed needs to be mentioned.
  11. It is interesting to know the luciferase activity, the authors need to look into downstream regulators of TGFb signalling pathway such as SMAD7, PAI-1 for SMAD2 signalling and SMAD6 for BMP signalling.
  12. It would be interesting to know the biological significance of the pathway. There are different types of EMT, can this signalling lead to upregulation of EMT markers?
  13. The schematic picture needs some modification, it is better to describe the R-SMADs in the picture and the downstream regulators. The triangles depicted in the picture is confusing.
  14. SB43, needs full abbreviation.
  15. The below references relate to Non-Smad signalling pathways.

The type I TGF-beta receptor engages TRAF6 to activate TAK1 in a receptor kinase-independent manner, Nature Cell Biology.

Non-Smad signaling pathways, Yabing Mu et al, Cell Tissue Res

Author Response

The article entitled ' Smad2/3 activation regulates Smad1/5/8 signalling via a negative feedback loop to inhibit 3T3-L1 adipogenesis’ by Aykul et al is very interesting and provides compelling evidence of the role of BMP signalling in adipogenesis.  The authors needs to address the below comments.

 Comments

  1. The article suggested that LDN-193189 inhibitor suppresses adipogenesis, by inhibiting the kinase activity of the BMP receptors, however, the LDN-193189 inhibitor also inhibits p38 signalling orchestrated by TGFbeta via the non-smad signalling cascade (TAK1-TRAF6-P38 cascade). Can the authors provide evidence that this is solely dependent on BMP signalling?

We thank the reviewer for pointing out this detail. While we do not have direct evidence that LDN is inhibiting p38 in this system, we already present indirect evidence. Namely, our phospho-proteome array (Figure S4, reference spots A3, A4) reveals that p38 is not phosphorylated at Thr180/Tyr182 (i.e., activated) in 3T3-L1 pre-adipocytes. Thus, we speculate that LDN targeting of p38 should not have a role in suppressing 3T3-L1 adipogenesis, and we maintain that the principal activity we observe for LDN, which manifests as adipogenesis inhibition, is inhibition of BMP receptor signaling.

  1. In Fig 5 A, B, C, D, E. The WB data needs total SMAD2/3 blots and the respective receptors (TBRII, ActRIIA etc) immunoblots. For D and E, total SMAD 1/5/8. Densitometry of WB data is required to interpret the data.

We have added the requested total SMAD immunoblots for 5A and B as supplemental figure S4 and S5.

With respect to receptors, we note that levels of endogenously expressed TGFb receptors are typically very hard to detect by Western Blot. This is generally recognized in the community. We nevertheless attempted to visualize BMPRII, TGFBR2, ActRIIA/B, ALK2 and ALK3 in the short time provided. We were able to obtain a WB signaling for BMPRII, ActRIIA/B, and ALK3, but not TGFBR2 and ALK2. We found that the levels of the detected receptors did not change appreciably with differentiation or with treatment (control vs. TGFBRII-Fc). WBs are shown in Supplemental Figure S6.

With respect to 5C, in this figure we screened different antibodies that probe both phospho and non-phospho SMAD forms and we examined the exact same sample we used in 5A. Thus, the requested total SMAD and loading controls are already shown either as part of the figure (SMAD vs p-SMAD), or as part of another panel (actin loading control, panels 5A/B).

With respect to 5D, this is an IP sample taken with the phospho-antibody and the WB is with different antibodies, including the non-phospho SMAD1 (S1), SMAD5 (S5), SMAD2 (S32) and SMAD3 (S3). With respect to 5E, the middle panel WB was probed with ana SMAD1 and the bottom panel was probed with a SMAD5 antibody. Again, the requested controls are already part of the figure. For 5E we treated one sample and loaded equal volumes. Thus, the amounts of protein are the same

We have added the densitometric analysis as supplemental figure S3. For 5E we have added the densitometric analysis in table form (table S4).

  1. The Mol size of the proteins needs to mentioned in the immunoblots.

We have fixed this oversight.

  1. Fig 6A, total SMAD 1/5, SMAD 2/3 is missing.

We have added total SMAD WB as supplemental figure S7. Consistent with our p-SMAD2/3 WB data, SMAD2 is in both nuclear and cytoplasmic fractions. SMAD1 appears more strongly in the cytoplasmic fraction.

  1. How do we differentiate localization of proteins in nuclear or cytoplasmic compartments? Can the authors probe the same samples for laminin A, a marker for nuclear fraction and tubulin, a marker for cytoplasmic fraction?

We have added Tubulin WB as supplemental figure S7. Tubulin is significantly reduced in the nuclear fraction as expected, indicating that fractions differentiate localization of proteins into nuclear or cytoplasmic compartments. We tested for Lamin B1, as that appears to be the most widely used nuclear marker, but could not detect this protein in our samples. We note that Lamins are not well detected in differentiating 3T3-L1 cells as Lamins inhibit 3T3-L1 differentiation (https://doi.org/10.1093/hmg/ddi480).

  1. Fig 6A, labelling of the samples in each well is misaligned, Please format it.

We have fixed the label misalignment.

  1. In Fig 6B, The Y axis information is missing (the time points correspond to?)

This may be a problem with formatting as units are shown in the figure (RLU). We have clarified what the time points correspond to in the legend.

  1. Fig 7, total SMAD 1/5 , SMAD 2/3 is missing.

We apologize that we could not carry out this control, as we did not have samples left over and could not grow cells and obtain new samples in a reasonable time frame for the resubmission. We note that it would take at least 4 weeks for the entire experiment to be completed, assuming every step works. Nevertheless, we point to other total SMAD figures, including S4 and S6, which indicate that SMADs are expressed and p-SMAD levels are independent from total SMAD. We further note that total SMAD levels are less important in this context as only p-SMAD are active and thus relevant to the biological question we are asking.

  1. In Fig 7, In the legend text, the conc of the inhibitor used is described in M. please clarify

We have fixed this formatting error.

  1. Number of experiments performed needs to be mentioned.

In the original submission we mention the number of experiments in the figure legend as follows: Statistical significance from four biological replicates was determined by two-way ANOVA and Tukey’s multiple comparison test.

Is this what the reviewer wanted?

  1. It is interesting to know the luciferase activity, the authors need to look into downstream regulators of TGFb signalling pathway such as SMAD7, PAI-1 for SMAD2 signalling and SMAD6 for BMP signalling.

We considered the reviewer’s suggestion and carried out a SMAD6 and SMAD7 WB analysis of various fractions (Figure S11). We found that SMAD6 is not detectable. By contrast, SMAD7 appears to accumulate as cells differentiate (more SMAD7 at day 8 than day 3). Notably, SMAD7 appears lower in cells treated with adipogenesis inhibitors and increased in cells treated with BMP-6 at day 8 of differentiation. However, as our day 3 data correspond to a more critical timepoint and results from this timepoint are less clear, we chose to refrain from speculation about the roles of SMAD7 in adipogenesis.

  1. It would be interesting to know the biological significance of the pathway. There are different types of EMT, can this signalling lead to upregulation of EMT markers?

We agree that it would be very interesting to find out the fate of these cells after SMAD2/3 activation. While it is tempting to speculate that SMAD2/3 activation induces EMT, we do not have data or the ability to measure EMT markers in the time provided for the review. We have nevertheless considered this possibility and may attempt to analyze the fate of these cells in future work.

The schematic picture needs some modification, it is better to describe the R-SMADs in the picture and the downstream regulators. The triangles depicted in the picture is confusing.

We modified the schematic by changing the SMAD depiction and labeling SMADs in the picture. We have attempted to simplify the diagram and altered the figure legend for clarity.

  1. SB43, needs full abbreviation.

We are not sure what the reviewer means by this. Would it be possible to clarify?

  1. The below references relate to Non-Smad signalling pathways.

We have added the suggested references to the expanded introduction.

The type I TGF-beta receptor engages TRAF6 to activate TAK1 in a receptor kinase-independent manner, Nature Cell Biology.

Non-Smad signaling pathways, Yabing Mu et al, Cell Tissue Res

Reviewer 2 Report

This paper explores the role of TGF-beta signaling in adipocyte differentiation in the well-known model of the murine cell line 3T3-L1. Authors treat cells with various TGF-b family ligands and Fc-fusion ligand traps, monitoring accumulation of lipids as a marker of adipogenesis.

They focus on ligand traps that inhibit differentiation, and analyze the activation of intracellular effectors SMAD1/5/8 and SMAD2/3. By performing phospho-specific immunoblots and luciferase reporter assays they find that molecules that prevent adipogenic differentiation cause inhibition of smad1/5 and activation of smad2/3.

They find that smad1/5 is hyperphosphorylated by some of the ligands, and such modification confines the protein outside the nucleus, presumably preventing transcription. Accordingly, a smad1/5 inhibitory drug prevents formation of lipid droplets, phenocopying the treatment with anti-adipogenic ligand traps.

Using small molecule inhibitors, they find that smad2/3 activation is required for the anti-adipogenic effect of ligand-traps, and inhibition of smad2/3 in fact prevents smad1/5 inhibition by anti-adipogenic ligands.

The results are convincing and potentially interesting, but I found the manuscript difficult to read, convoluted, at times even confusing; in other words, I had a hard time following the “story” and grasping the rationale behind the various experimental steps.

For instance, the introduction is heavily unbalanced. More than half of it (lines 40-56) is in fact a summary of the results, and can be removed. Instead, the introduction should be expanded to better describe the model and what is known on TGF signaling in adipogenesis, and to provide the reader with the conceptual tools necessary to understand and appreciate the logic and the implications of the subsequent experiments.

Below the issues that I think need to be addressed.

-Experiments show the same effect when cells are treated with TGFb1 and TGFbRII-Fc (the corresponding ligand trap). Idem with mCryptic-Fc: it binds Activin B, but has the same effect as Activin B on the cells. How can the authors explain the same phenotype upon treatment with a ligand or its inhibitor ligand trap?

I think this is kind of troubling, and I was not satisfied with the superficial comment in the discussion stating that “…Our results, therefore, suggest that an interplay of multiple ligands may be required to direct 3T3-L1 cells and other precursors toward adipogenic fates…”, because the effects on lipogenesis are observed with single purified ligands or ligand traps. I’m aware of the complexity of the model, and the potential crosstalk of different pathways, but this contradictory observation cannot be ignored.

-The authors use in various assays alternatively TGFbRII-Fc or TGFb1, or other ligand traps with apparently no rationale. This makes it difficult to actually compare the downstream events. For instance, why the RT-PCR analysis of adipogenic target genes was done with mCryptic-Fc and Cripto1-Fc, while most other experiments were done with TGFbRII-Fc ??

-It may be useful to complement the data obtained with the BRE-LUC reporter with a co-IP showing whether SMAD1/5 binds SMAD4 when hyperphosphorylated after TGFbRII-Fc treatment.

Figure S1 - mRNA levels don’t always correlate with protein levels; it is possible that some receptors or ligands are expressed at different levels than those that are assumed based on the microarray data extracted from GSE 20696. Maybe that could partially explain some of the apparently contradictory observations?

Figures 1-3. It may be useful to normalize the number of lipid droplets for the number of cells, as LD/nuclei/well

Figure 3. Why only this figure has Nile red staining? What’s its significance? Legend probably has an error between panels C and D.

Figure 4. legend mentions colors but figure is grayscale

Figure 5A,B. Western blots lack the cells at time zero of differentiation, that would be nice as a baseline control. They also lack a pan-SMAD staining, that would allow monitoring possible changes in the levels of SMAD proteins.

Figure 5C. These blots lacks loading control and Mw. Also, to compare hi and lo bands, the ALK3-Fc treated and vehicle control should be run side by side on the same gel.

Furthermore: it appears that two SMAD5-specific antibodies (Cs and Ab) label different bands. Explanation?

Figure 7. It seems that the inhibitor LDN induces the shift of p-SMAD1/5 to the hyper-phosphorylated slow migrating form even in untreated cells. The signal is low, but the migration shift is clear. I wonder if this migration shift has been already documented upon LDN treatment; perhaps LDN-mediated inhibition of the canonical SMAD1/5 activating kinases may “uncover” the still unknown kinase responsible for the shift-inducing phosphorylation? In fact, LDN does not prevent hyper-phosphorylation of SMAD1/5 when cells are treated with the TGFbRII-Fc ligand trap.

Author Response

This paper explores the role of TGF-beta signaling in adipocyte differentiation in the well-known model of the murine cell line 3T3-L1. Authors treat cells with various TGF-b family ligands and Fc-fusion ligand traps, monitoring accumulation of lipids as a marker of adipogenesis.

They focus on ligand traps that inhibit differentiation, and analyze the activation of intracellular effectors SMAD1/5/8 and SMAD2/3. By performing phospho-specific immunoblots and luciferase reporter assays they find that molecules that prevent adipogenic differentiation cause inhibition of smad1/5 and activation of smad2/3.

They find that smad1/5 is hyperphosphorylated by some of the ligands, and such modification confines the protein outside the nucleus, presumably preventing transcription. Accordingly, a smad1/5 inhibitory drug prevents formation of lipid droplets, phenocopying the treatment with anti-adipogenic ligand traps.

Using small molecule inhibitors, they find that smad2/3 activation is required for the anti-adipogenic effect of ligand-traps, and inhibition of smad2/3 in fact prevents smad1/5 inhibition by anti-adipogenic ligands.

The results are convincing and potentially interesting, but I found the manuscript difficult to read, convoluted, at times even confusing; in other words, I had a hard time following the “story” and grasping the rationale behind the various experimental steps.

For instance, the introduction is heavily unbalanced. More than half of it (lines 40-56) is in fact a summary of the results, and can be removed. Instead, the introduction should be expanded to better describe the model and what is known on TGF signaling in adipogenesis, and to provide the reader with the conceptual tools necessary to understand and appreciate the logic and the implications of the subsequent experiments.

We thank the reviewer for the constructive suggestion and expanded the introduction by adding information on the general problem of adipogenesis and the specific roles of tgf signaling in this process.

Below the issues that I think need to be addressed.

-Experiments show the same effect when cells are treated with TGFb1 and TGFbRII-Fc (the corresponding ligand trap). Idem with mCryptic-Fc: it binds Activin B, but has the same effect as Activin B on the cells. How can the authors explain the same phenotype upon treatment with a ligand or its inhibitor ligand trap?

We agree that this is a puzzling observation. Nevertheless, it represents a real result that is highly reproducible. We speculate that TGFbRII-Fc may trap high affinity ligands and alter a potential squelching activity. Thus, as observed in transcription, a strong ligand could also act as signaling inhibitor or antagonist. Several recent publications discuss this phenomenon (PMID: 28886385, 33087301, 26961869). We note that TGFbRII-Fc modulates biology in a different way than the small molecule kinase inhibitor (SB43), which simply inhibits canonical SMAD2/3 activating kinases and rescues the effect on adipogenesis of both TGFb1 and TGFBR2-Fc. We have expanded on this point in the discussion.

I think this is kind of troubling, and I was not satisfied with the superficial comment in the discussion stating that “…Our results, therefore, suggest that an interplay of multiple ligands may be required to direct 3T3-L1 cells and other precursors toward adipogenic fates…”, because the effects on lipogenesis are observed with single purified ligands or ligand traps. I’m aware of the complexity of the model, and the potential crosstalk of different pathways, but this contradictory observation cannot be ignored.

As mentioned above, recent manuscripts have demonstrated that ligands can be activating and inhibiting, that cells perceive tgfb/bmp signals in combinatorial fashion. We expand on this point as noted above.

-The authors use in various assays alternatively TGFbRII-Fc or TGFb1, or other ligand traps with apparently no rationale. This makes it difficult to actually compare the downstream events. For instance, why the RT-PCR analysis of adipogenic target genes was done with mCryptic-Fc and Cripto1-Fc, while most other experiments were done with TGFbRII-Fc ??

We agree with the reviewer’s sentiment. The reason for using various traps is that the project developed from a screen to identify traps with anti-adipogenic activity. At first, we carried out experiments with traps that had better IP potential. As we probed the mechanism, we switched to TGFbRII-Fc because we can produce it more easily at the levels required for this type of work and because it is more stable. Hence the variable use. Nonetheless, based on the two very strong and distinct phenotypes that we observe, and the comparable biochemical behavior of either phenotype, we have very strong evidence indicating that all anti-adipogenic traps and ligands engage a similar mechanism.

-It may be useful to complement the data obtained with the BRE-LUC reporter with a co-IP showing whether SMAD1/5 binds SMAD4 when hyperphosphorylated after TGFbRII-Fc treatment.

We appreciate this suggestion but would like to note that co-IP with endogenous SMAD proteins is very challenging due to their low abundance. In fact, we attempted many times to IP activated (i.e. phospho-) SMAD and identify it’s phosphorylation status with Mass Spec. We found that the assay was not sensitive enough to detect the ph. In figure 5D, the reviewer may appreciate how weakly the SMAD1 and SMAD5 Abs react with the IPed p-SMAD1/5/8. We hope the reviewer understands the technical difficulty of this problem and appreciates our efforts, which, unfortunately, did not yield useful results.

Figure S1 - mRNA levels don’t always correlate with protein levels; it is possible that some receptors or ligands are expressed at different levels than those that are assumed based on the microarray data extracted from GSE 20696. Maybe that could partially explain some of the apparently contradictory observations?

We agree with the reviewer that mRNA levels don’t always correlate with protein levels. We note that we analyzed several microarray datasets with more or less the same conclusions. While we would hope that mRNA data could shed light on the squelching or interplay mechanism, mining of mRNA data will, unfortunately, not be sufficient to explain our observations.

Figures 1-3. It may be useful to normalize the number of lipid droplets for the number of cells, as LD/nuclei/well

We changed the figure as suggested and recalculated the statistical significance.

Figure 3. Why only this figure has Nile red staining? What’s its significance? Legend probably has an error between panels C and D.

We apologize for causing this confusion. Our reason for adding a fourth panel was purely cosmetic. We have adjusted the figure to match figures 1 and 2 by decreasing the size of the panels.

We fixed the figure legend.

Figure 4. legend mentions colors but figure is grayscale

We fixed this oversight.

Figure 5A,B. Western blots lack the cells at time zero of differentiation, that would be nice as a baseline control. They also lack a pan-SMAD staining, that would allow monitoring possible changes in the levels of SMAD proteins.

We now show the time point in 5A (same as 5B) taken at time 0, which we had omitted from the figure previously.

We also added a SMAD analysis as Supplemental Figures S4 and S5.

Figure 5C. These blots lacks loading control and Mw. Also, to compare hi and lo bands, the ALK3-Fc treated and vehicle control should be run side by side on the same gel.

We added the Mw values to 5C.

The requested loading controls are already present in figure 5A, as figure 5C consists of exactly the same samples as in 5A. To generate figure 5C, we just carried out WBs using anti p-SMAD1/5, anti-pSMAD2/3 as controls (as we show in 5A) and loaded the same amounts of ALK3-Fc/Veh sample as in 5A.

With respect to running ALK3-Fc treated and vehicle control side by side on the same gel, we appreciate that this would have been the better approach. Unfortunately, we do not have sufficient sample or time to repeat this experiment in the suggested way. To improve clarity, we labeled Mw weight markers in the file showing the entire WBs and aligned the two WBs accordingly. The hope this additional insert provides a clearer indication of electrophoretic mobility.

Furthermore: it appears that two SMAD5-specific antibodies (Cs and Ab) label different bands. Explanation?

We are as puzzled as the reviewer by this result. However, we speculate that the observed differences could reflect differences in the antigenic sequence recognized by each MAb. The cell signaling antibody was raised against a “synthetic peptide corresponding to residues surrounding Pro249 of human Smad5”. The Abcam antibody was raised against a “synthetic peptide within Human SMAD5 aa 200-300”. Both sequences form part of the SMAD linker region, which can be post-translationally modified by phosphorylation (See supplemental figure S3). As the antigenic regions recognized by the two antibodies may be different, it is conceivable that the antibodies are differentially affected by linker phosphorylation. In addition, the Abcam antibody may bind its antigen with higher affinity than the cell signaling antibody and thus may have given a higher signal for the unphosphorylated form.

Figure 7. It seems that the inhibitor LDN induces the shift of p-SMAD1/5 to the hyper-phosphorylated slow migrating form even in untreated cells. The signal is low, but the migration shift is clear. I wonder if this migration shift has been already documented upon LDN treatment; perhaps LDN-mediated inhibition of the canonical SMAD1/5 activating kinases may “uncover” the still unknown kinase responsible for the shift-inducing phosphorylation? In fact, LDN does not prevent hyper-phosphorylation of SMAD1/5 when cells are treated with the TGFbRII-Fc ligand trap.

We appreciate the reviewer’s observation. We also noted this change but did not discuss thoroughly in an attempt to keep the manuscript streamlined. To our knowledge, a p-SMAD1/5/8 shift caused by LDN has not been documented. However, p-SMAD1/5/8 bands of different molecular weights have been widely observed. One report that discussed the nature of the various p-SMAD1/5/8 forms is Daly et al. 2008, cited here.

We agree with the reviewer that this observation could point to a non-canonical kinase in the shift-inducing phosphorylation. We have expanded this point by altering the corresponding paragraph as follows:

Several kinases, including ERK, JNK and PI3K, are thought to hyper-phosphorylate SMADs [62-69]. To determine if any one of these kinases engages in SMAD1/5/8 hyperphosphorylation, we investigated their activation state using the phospho-kinase profiler array kit (Figure S8). All tested samples, including undifferentiated precursors and differentiating adipocytes, showed significant GSK-3α/β and WNK1 phosphorylation, suggesting that these kinases may have relevant roles in adipogenesis. However, these results also indicate that GSK-3α/β and WNK1 may not be involved in 3T3-L1 adipogenesis arrest, as their activation levels are unchanged in treated cells. By contrast, ERK phosphorylation was increased in cells treated with adipogenesis inhibitors. However, further analysis did not provide evidence that ERK gives rise to p-SMAD1/5Hi. Taken together, these results indicate that a still unknown kinase may be responsible for the proposed, shift-inducing phosphorylation.

Round 2

Reviewer 2 Report

I am satisfied with the additional data included in this revised version of the paper, and the explanation provided to some of my perplexities. I think the manuscript is acceptable for publication.